# Temperature-Dependent Effects of Induced Hyperthermia, Including Whole-Body Hyperthermia, on the Hallmarks of Cancer: A Systematic Review

**DOI:** 10.3390/cancers17233824

**Published:** 2025-11-28

**Authors:** Ivana Gorbaslieva, Tom Quisenaerts, Johannes J. P. M. Bogers, Marc Peeters, Vera Saldien, Dirk Ysebaert

**Affiliations:** 1Faculty of Medicine and Health Sciences, University of Antwerp, 2610 Wilrijk, Belgium; 2Department of Hepatobiliary, Transplantation and Endocrine Surgery, University Hospital of Antwerp, 2650 Edegem, Belgium; 3Department of Anesthesiology, University Hospital of Antwerp, 2650 Edegem, Belgium

**Keywords:** induced hyperthermia, hallmarks of cancer, immune activation, genomic instability, temperature-dependent effects, systematic review

## Abstract

Cancer remains one of the most complex diseases to understand and treat because it involves many biological processes that allow tumors to grow and spread. One promising supportive treatment is induced hyperthermia, which carefully raises the body’s temperature to weaken cancer cells and strengthen the body’s natural defenses. This study systematically reviewed existing research to understand how hyperthermia affects the main biological features that define cancer, known as the hallmarks of cancer. The findings show that heating tumors within specific temperature ranges can stimulate the immune system and make cancer cells more sensitive to treatment. However, at very high temperatures, the cell-killing effects become indiscriminate. By summarizing these mechanisms, this research helps clarify how temperature-based therapies could be safely combined with other treatments, guiding future studies to improve cancer therapy effectiveness.

## 1. Introduction

The hallmarks of cancer were proposed as a set of functional capabilities acquired by human cells as they evolve progressively to a neoplastic state. These specific capabilities are crucial for their ability to form malignant tumors.

The ten hallmarks include the acquired capabilities for 1. avoiding immune destruction; 2. genome instability and mutation; 3. resisting cell death; 4. deregulating cellular metabolism; 5. inducing/accessing the vasculature; 6. enabling replicative immortality; 7. sustaining proliferative signaling; 8. evading growth suppressors; 9. tumor-promoting inflammation; and 10. activating invasion and metastasis [1].

Induced hyperthermia (also referred to in this review as hyperthermia or HT) is the procedure of increasing the intended temperature of a part or the entire body above the normal setpoint for a defined period. It is applied alone or as an adjunctive treatment to various types of cancer [2]. In contrast to local or regional hyperthermia, whole-body hyperthermia (WBHT), or systemic induced hyperthermia, represents the only hyperthermia modality inducing certain desired effects for locally advanced cancer and patients with metastatic cancer [3].

The first results of long-term local/regional heating (48 h at 42–44 °C) to destroy tumors without damaging healthy tissues were reported in 1898. In 1916, reports described the 3–7-year survival of patients with inoperable cancer of the uterus after local hyperthermia above 45 °C, and Behrouzkia et al. confirmed these results [4].

The first paper on local and whole-body hyperthermia (WBHT) combinations was published in 1977 [4]. Since then, a wide range of whole-body hyperthermia application methods has been proposed and tested, including patients submersion in hot wax or liquid, encasement in a high-flow water perfusion suit, wrapping patients in plastic, and the use of extracorporeal perfusion [5]. WBHT has also been used as a hot air model-based system [6]. However, whole-body hyperthermia has not been implemented in clinics yet because of the uncontrolled method of intervention, leading to a lack of safety.

The biological rationale for the treatment of cancer by heat is based on the difference in its effect on healthy/immune/tumor cells, which is dependent on the temperature and duration of heating [7]. In addition, the tumor cell environment is beneficially influenced by hyperthermia. The differential sensitivity of healthy and tumor cells to heat is dependent on the cell type and cell environmental conditions, whereas hyperthermia enhances the biological effects of both radiation and chemotherapy [8,9].

The knowledge gap we aim to address is understanding the potential mode of action of hyperthermia in cancer, considering the temperature, duration, cell interactions, and pathway activation/inhibition. This systematic review provides a synthesized, comprehensive insight into hyperthermia by examining its impacts through the perspective of the hallmarks of cancer. This methodology sets it apart from the current literature, which mainly emphasizes a single mode of action or the clinical results of different types of cancer. By improving our comprehension of the biological processes affected by hyperthermia, based on multiple mechanisms of action—categorized through the Hallmarks of Cancer framework—this review establishes a broader understanding of its biological impact and establishes a rationale for upcoming studies and the creation of more targeted and efficient cancer treatments.

## 2. Materials and Methods

A systematic search of the available literature was conducted in accordance with the principles of the PRISMA (Preferred Reporting Items for Systematic Reviews and Meta-Analyses) framework and Cochrane guidelines to identify eligible studies investigating the effects of induced hyperthermia on the hallmarks of cancer. This review has not been registered. Literature searches were performed in Medline (PubMed) and the Cochrane Library for the period from 1 January 2000 to 17 June 2025. The search strategy combined MeSH terms and publication types related to neoplasms and induced hyperthermia (excluding malignant hyperthermia), limited to systematic reviews and reviews.

### 2.1. Eligibility Criteria

The following inclusion criteria were used to select the literature: (I) papers published after the 1st of January 2000 focused on mode-of-action results covering fundamental research published earlier rather than efficacy data; (II) a review or systematic review; (III) focused on mode-of-action data about induced hyperthermia; (IV) focused on neoplastic disease and cellular responses to tumors in vivo, cancerous cells in vitro; (V) based on research using mammalian cells; (VI) reporting on HT in the fever range to mild hyperthermia (38–45 °C). Exclusion criteria consisted of the following aspects: (I) articles published before the 1st of January 2000; (II) papers on mechanisms of noncancer disease; III) malignant hyperthermia; (IV) efficacy data; (V) prokaryotic cells or nonmammalian eukaryotic cells; (VI) hypothermia to normothermia (36–38 °C); (VII) microwave ablation (≥45 °C); (VIII) techniques of assessment, intervention or otherwise, which might confound with the observed effect (e.g., LPS or bacterial contamination); and (IX) hyperthermia techniques implying other effects (induction of infection, PDT, HIFU, and electro hyperthermia).

Systematic reviews and reviews were included instead of primary studies to allow for a comprehensive synthesis of mechanistic insights across the hallmarks of cancer, integrating evidence from high-level, peer-reviewed sources.

### 2.2. Search Strategy and Databases

A structured literature search was performed in Medline (PubMed) using Boolean operators (AND, OR, NOT) and extended to the Cochrane Library databases. The following fixed query served as the general search template for PubMed: “(((“Neoplasms”[MeSH Terms]) AND (“Hyperthermia, Induced”[MeSH Terms] NOT “Malignant Hyperthermia”[MeSH Terms])) AND (“Systematic Review”[Publication Type] OR “Review”[Publication Type] OR “Meta Analysis”[Publication Type])) AND (2000/01/01:2025/06/17[Date–Publication])”. This query was further refined with additional PICO-specific keywords to capture literature addressing specific cancer types and underlying mechanisms. To complement this, a broad search was conducted in the Cochrane Library using the following query: “MeSH descriptor: [Hyperthermia, Induced] + MeSH descriptor: [Neoplasms]”.

Articles underwent a primary appraisal based on titles and abstracts; those that clearly did not meet predefined inclusion/exclusion criteria were excluded and recorded separately. For articles retained after this stage, full texts were obtained via institutional journal access or open access sources. These articles then underwent a secondary appraisal using the full texts, with each article evaluated against the predefined inclusion/exclusion criteria. All included and excluded studies were documented, with explicit reasons provided for exclusions. Summaries were prepared for analysis. To enhance reliability and reduce bias, all retained articles were independently assessed by two teams of the listed authors. Each team evaluated the articles independently of the other. In cases of discordance, an additional author decided whether to include or exclude the article. Systematic reviews were graded using the PRISMA checklist (Appendix A), and all eligible papers were catalogued in Mendeley.

## 3. Results

A total of 2015 records were electronically found, and 103 papers were ultimately included for data extraction, as listed in Figure 1. After the initial screening and review, where studies without relevant keywords in the title or abstract were excluded, 179 articles were retained. Subsequent full-text review resulted in 103 articles being included in the final review, with 73 post hoc exclusions due to lack of topical relevance.

Figure 2 presents a summarized overview of how induced hyperthermia affects the hallmarks of cancer, focusing on the most relevant to its mechanism of action and grouped into temperature intervals from 39 °C to 43 °C.

The results of this systematic review are discussed in separate sections, per relevant hallmark of cancer, and summarized in Figure 3.

### 3.1. Avoiding Immune Destruction

#### 3.1.1. Induced Hyperthermia Triggers Immune-Mediated Cancer Destruction

Under physiological conditions, the immune system plays a significant role in preventing tumor formation and progression. Importantly, functional deficiencies in CD8+ cytotoxic T lymphocytes (CTLs), CD4+ Th1 helper T cells, or natural killer (NK) cells are associated with increased tumor incidence [10,11]. For example, patients with colon and ovarian cancers tend to have a more favorable prognosis when their tumors show high levels of lymphocyte infiltration, particularly by cytotoxic T cells (CTLs) and natural killer (NK) cells, compared to patients whose tumors lack such immune cell presence [12,13]. However, cancer cells may disable infiltrating CTLs and NK cells by secreting TGF-β or other immunosuppressive factors [14].

Various research groups have demonstrated that induced hyperthermia influences the immune system in the cancerous stage, showing immune activation around 39 °C, while at higher temperatures (above 41.5–42 °C), it may lead to immune inhibition or suppression [15,16].

In terms of immune cells, fever-range induced hyperthermia extends the overall survival time of lymphocytes. Fever-range hyperthermia increases the expression of c-FLIP, a caspase-8 inhibitor, whereas caspase-8 itself is crucial for lymphocyte survival and thus for sustaining an activated immune response [17]. The potential diminished activity of lymphocytes is transient and only present at relatively high temperatures [18]. There is a correlation between the extent of immune cell infiltration (e.g., post-HT treatment) in cancer patients and the survival rates of immune cells, indicating the potential gain of immune activation and tumor infiltration by immune cells after HT treatment [19,20].

#### 3.1.2. Central Role of Heat Shock Proteins (HSPs)

The release of both HSP70 and HMGB1, which are considered danger signals or danger-associated molecular patterns (DAMPs), is significantly increased when HT is combined with radiotherapy (RT) [21]. The increased release could be correlated with increased necrotic cell death or thermally stressed cells. HT and RT work synergistically, which in turn leads to immune activation [15,17,22,23,24,25,26,27]. However, HMGB1 expression in tumor cells promotes inflammation, angiogenesis, evasion of cell death, and survival of myeloid-derived suppressor cells (MDSCs) through the induction of autophagy and metastasis [15,27]. The large fluctuations in the extracellular levels of HMGB1 induced by hyperthermia lead to its antitumor effects instead of its chronic (low-level) protumor effects [27]. Induced hyperthermia is a potent activator of the release and presentation of HSPs by tumor cells, whereas radiotherapy is capable of inducing HSP presentation but requires the addition of hyperthermia to result in HSP shedding [17,18,23,25,28,29,30,31,32,33]. This review clearly highlights the central role of HSPs in hyperthermia-induced immune activation. At normothermic temperatures, cancer cells often express HSPs, whereas healthy cells often lack this expression. Induced hyperthermia increases cancer cell HSP expression, and their release/membrane presentation acts as a danger signal to neighbouring immune cells, causing their activation [15,16,18,23,24,25,27,28,29,30,31,34,35,36,37,38,39,40,41,42,43]. HSPs can be presented or recognized by receptors on cell membranes; HSP60, by CD14; HSP70, by CD14, CD40, CD91, or Lox-1 receptors; gp96, by CD91 (*α*-2 macroglobulin/LDL receptor-related protein); and SR-A, HSP90, by CD91, and HSPs, in general, by CD40. This binding/recognition leads to the release of proinflammatory cytokines [15,17,18,28,29,39]. Some data have shown that CD91 can be used by APCs to internalize HSPs (gp96, HSP90, and HSP70) [17,29]. HSPs and their chaperoning specific epitopes are capable of triggering both signal 1 (MHC upregulation) and signal 2 (costimulatory mediator upregulation). While the presence of HSP70 on the cell surface leads to the activation of NK cells, it also protects cells from radiation-induced damage due to its intercellular presence [16]. How the shedding of HSPs occurs is not yet fully understood. The possible mode of action is achieved through its binding to globotriaosylceramide (Gb3), which resides in the cell membrane and acts as a flip-flop mechanism to present or shed HSPs [15,31]. Phosphatidylserine (PS) is described as an anchoring point for HSPs. PS is located at the outer cell membrane and is also an early indicator of apoptosis. Cells expressing high levels of PS-HSP70 complexes are highly resistant to cellular stress [42]. This resistance may be related to thermotolerance induced by high levels of HSPs, which possibly upregulate surface expression and prohibit the induction of apoptosis. Thermotolerance refers to a temporary resistance to heat that develops after exposure to a prior heat treatment [44]. In the case of high HSP70 expression by tumor cells, the additional margin of increase after hyperthermia exposure is rather limited [17,30,37,42]. HSPs play an important role in activating the adaptive immune system. Their role as specialized carriers focuses on presenting chaperoned proteins to immune cells, including tumor-specific mutated proteins that are not recognized as ‘self’ epitopes. This function of HSPs enhances the presentation of potential exogenous epitopes to the MHC-II molecule but also activates cross-presentation pathways to MHC-I [15,16,17,18,24,25,27,28,29,30,31,33,35,37,38,39,40,41,45,46,47,48,49,50,51,52,53,54,55].

In detail, HSP70 and 90 transport the epitope to the endoplasmic reticulum (ER) through a transporter associated with antigen presentation (TAP). In the ER, the epitope binds to gp96, which in turn transfers the epitopes to the MHC-I β2-microglobulin complexes (Figure 4a). In this process, HSP70 expression peaks at 24 h after HT, whereas HT induced MHC-I activity peaks at 48 h [28]. HT upregulates ubiquitin, which is an important component for the presentation of tumor-specific epitopes [17]. HSP chaperones are degraded by proteasomes and TAP28, which results in peptides that can be presented to MHC-II (Figure 4a). In addition, a vacuolar pathway of degradation leads to the presentation of exogenous peptides by MHC-I [29]. 29HT also increases the processing rate of tumor-specific antigens by upregulating the immunoproteasome (the Imp2 and Imp7 subunits) [17]. Owing to increased affinity but not increased expression, MHC-I and MHC-II are upregulated in the mild range of HT or 43 °C [17,18,28,29,36,45]. In contrast, other evidence revealed no difference in MHC-I expression below 42.5 °C and a decrease above 42.5 °C [31,37,45]. A lack of MHC-I is frequently associated with metastatic tumors [42]. This evasion of CTLs is counteracted by the loss of NK cells. However, signals are still needed to activate an NK response, known as the ‘missing self’ hypothesis [31,37,42,46]. No changes are reported by HSP70-induced MHCI, II, or specific immunogenicity at 42.5 °C [37,42,46].

In summary, the pivotal role of HSPs in modulating the immune-stimulating effects of induced hyperthermia is temperature dependent. Under elevated temperatures, cancer cells upregulate and release HSPs, which act as danger signals to activate innate and adaptive immune responses. Additionally, thermotolerance—driven by either a high basal HSP expression in tumors or after previous hyperthermia exposure—can limit further immunogenic effects of repeated induced hyperthermia, suggesting a potential reduction of therapeutic benefit.

#### 3.1.3. Antigen-Presenting Cells

Induced hyperthermia increases the antigen presentation activity of APCs. Specifically, dendritic cells and macrophages are upregulated due to direct heat exposure or through activation by exposure to danger signals, such as HSP70 and HSP90, resulting in increased tumor-specific antigen presentation [17,18,23,28,30,38,42,45,56,57]. Independent of its chaperone function with tumor epitopes, HSP70 is also a potent activator of immune cells, prompting them to release cytokines such as TNF-*α* and thereby priming dendritic cells and macrophages/monocytes for APC functionality in an autologous positive feedback loop of TNF-*α* stimulation [23,28,30,39].

##### Dendritic Cells (DCs)

DCs form the bridge between the innate and adaptive immune systems, as they form an essential link and induce specific CTLs against cancer cells. This activation of DCs leads to increased APC function, the release of proinflammatory cytokines, and T-cell activation and migration [15,17,18,22,23,24,28,30,36,38,39,40,41,43,45,52,53,57,58,59]. Increased T-cell activation is caused by increased DC–T-cell synapse formation, which promotes the release of the cytokines TNF and IL-12 [38,51,53].

The activation of TLR2/TLR4 by HSPs causes the activation of HSF1 signaling followed by NF-kB signaling in DCs, resulting in the release of cytokines (IL-1, IL-6, and TNF) and the release of tumor antigens, which can be processed by APCs (Figure 4a) [17,18,19,23,28,29,38,39,45,53]. The presentation of tumor-specific epitopes to DCs or macrophages triggers the release of inflammatory cytokines, such as IL-1β, IL-6, IL-12, and TNF-*α*, and the upregulation of CD80/86, which induces the formation of specific Th-1 cells and DC maturation [18,29,36]. The exposure of DCs to HSPs (HSP60, 70, 73, 90, and gp96), induced by HT, can trigger DC maturation, increasing MHC-II expression and costimulatory signal expression (CD86) [15,23,30]. The degree of DC maturation is proportional to the level of HSP, whereas HSP absence/inhibition is associated with DC dysfunction [15,17,27,29,30,37,54]. HSP90-activated DCs are also responsible for IFN-⍺ release, which stimulates NK cell activation [22,46]. In addition, increased HSPs have been proven to be capable of increasing CTL levels in a CD4+ Th-cell-independent manner [29]. DC migration to lymphoid organs is regulated by increased activation of CCR7 and decreased activation of CCR6 after hyperthermia exposure [15]. Hyperthermia-activated DCs exhibit elevated expression levels of the IGFBP-6 gene. In addition, IGFBP-6 also induces metastasis and acts as a homing signal for circulating tumor cells [59]. Adding HT to RT synergistically enhances DC infiltration into solid tumors [22,60].

##### Macrophages

In addition to DC activation, induced hyperthermia also activates macrophages via HSF1, enhancing their function by 40% at temperatures up to 40 °C [15,17,18,23,45,59,60]. HSP90 causes an increase in iNOS activity in macrophages (producing NO), ROS, and NF-kB signaling (due to increased I-kB-⍺ degradation) and the upregulation of TLR4 [17,23,41]. In addition, macrophages are also activated by HSP23, 27, 40, and 70. HSP70 activates TAM macrophages, leading to the release of NO (increasing vascular permeability), TNF-⍺, and IL-6. These cytokines increase the release of CXCL1, 2, 10, endothelial ICAM-1, and VCAM-1, causing immune cell trafficking [23,41]. Hyperthemia upregulates CD14 and CD11b in monocytes. When activated, these monocytes infiltrate the tumor microenvironment [16].

#### 3.1.4. Effector Cells

##### NK Cells

When exposed to fever-range hyperthermia (peak activation at 39 °C), NK cells infiltrate and accumulate in tumor tissue at much higher rates, causing tumor regression. However, higher temperatures have an increasingly detrimental effect on NK cell activity [15,16,17,18,22,23,24,26,27,30,31,35,36,37,41,42,45,46,57,58]. NK cell killing activity is tightly regulated by a balance of pro- and anticytotoxic factors present on both the NK cells and the target cells. NKG2D is an important procytotoxic receptor on the NK cell surface that causes the activation of NK cell antitumor cytotoxicity. NKG2D functionality is increased by hyperthermia due to increased affinity, which is achieved by its clustering on the cell membrane [15,17,18,22,23,31,42,45,46,53,58]. NKG2D on CD8+ T cells can act as a costimulatory player to overcome TCR class I-restricted cytotoxicity [52,58]. In addition, MHC-I-related chain A (MICA), the ligand of NKG2D, is upregulated by hyperthermia-induced affinity or expression in the presence of its heat shock response element (HSE) [17,22,23,30,36,40,45,46,52,53,58,61]. In addition to its direct induced hyperthermia effect, HSP70 increases the activity of NK cells by binding to the CD94/NKG2D receptor superfamily, resulting in NK activation and proliferation (Figure 4b) [15,27,39,41,53]. HSP70, 72, 90, gp96, and HSF1 increase the NK cell killing capacity by stimulating the proliferation and upregulation of NKG2D, CD94, NKp46, and CD56, causing an increase in cell-cell adhesion on the NK cell membrane [17,23,31,39,46]. On the other hand, induced hyperthermia upregulates HLA-E expression by tumor cells, which stimulates the inhibitory NK surface signal CD94. However, in the presence of HSP60, it becomes masked, thus stopping its inhibitory effects [15,17,37].

##### T Cells

At fever-range hyperthermia (peak-activation at 39 °C), the activity of CD8+ CTLs increases, as does their responsiveness to cancer cells. Higher temperatures induce T-cell apoptosis [15,16,17,18,23,24,28,36,38,41,42,58,59]. Fever-range hyperthermia also activates (1) CD4+ Th cells, causing the release of IL-2 and further stimulating CTL development, and (2) potentiates the capacity of IL-1 to induce T-cell proliferation at temperatures up to 39 °C (Figure 4b) [38,42,53,58]. In contrast, prolonged CTL activation against tumor cells has been observed at 41.8 °C during short exposures (15–20 min) [15]. The cytotoxic functions of CTLs, such as granzyme B, perforin, IFN-γ, FasL (CD95L), and TNF-⍺, are elevated by induced hyperthermia, as is Fas receptor presentation on tumor cells (Figure 4b) [15,17,23,31,38,41,42,52,62]. HSF-1 is responsible for the upregulation of Fas-L in CTLs through increased NF-kB signaling [23,53]. HSP90 is involved in the upregulation of TNF-⍺ expression, the NF-kB signaling pathway, and the proinflammatory activation of CTLs [23,41]. In addition, induced hyperthermia decreases the number of CD4+ T cells in the blood while causing a slight increase in the number of CD8+ T cells due to increased production and extravasation. HSP70 induces an increase in IFN-*γ* and TNF-⍺ during the first 24 h after hyperthermia exposure and is associated with a marked increase in peripheral CTL and CD56 [18,62]. In addition, the levels of CTLs, CD56, sIL-2R, and lymphocytes expressing CD69 peak at 48 h after hyperthermia exposure [18]. As described, a certain ratio of T cells to target (tumor) cells must be achieved to induce tumor regression via T-cell immunity. The reported ratios vary from 20:1 to 1:1 for T cells and tumor cells, respectively [20]. Thus, increased infiltration of T cells by induced hyperthermia can help shift the ratio to achieve more effective levels.

##### B Cells

Induced hyperthermia-activated B cells express more TLR-9 receptors via the ERK and NF-kB signaling pathways (Figure 4b). HSP90 and HSF-1 promote the activation and proliferation of B cells [23,29,41]. HSP60 triggers the TLR4 receptor, protects B cells from apoptosis, and subsequently increases their survival time [23,41]. A member of the HSP110 family, Apg-2, induces B-cell proliferation while lowering oxidative stress [23]. HSP72 upregulates B-cell MHC expression [29,38]. The highest level of antibody secretion is reached between 39 °C and 40 °C and is T-cell-dependent, suggesting the involvement of T-helper cells in this hyperthermia-induced B-cell activation process [18].

##### Neutrophils

Up to 40 °C, neutrophil infiltration is increased in tumors, causing tumor regression due to hyperthermia-mediated antitumor immune effects. The proliferation and recovery of neutrophils are promoted by the release of IL-17, G-CSF, and IL-1 (Figure 4b) [57].

#### 3.1.5. Cytokines

Induced hyperthermia causes a high level release of proinflammatory cytokines and antibodies [18,23]. The antitumor toxicity of TNF⍺/β is significantly enhanced at 40 °C [18,62]. Increases in the serum levels of IL-1, IL-1β, IL-2, IL-6, IL-8, IL-10, G-CSF, GM-CSF, TGF-β, IFN-⍺, and TNF-⍺ were measured one hour after treatment at 41.8 °C [16,18,53,62]. In addition, bone marrow cells increase the release of IL-1, IL-6, and TNF-⍺. However, the peak concentration is reached after 4 cycles of induced hyperthermia [18]. In turn, IL-2, 7, 12, 15, and type 1 IFNs are known to rapidly activate lymphokine-activated killer (LAK) cells [46,62]. Hyperthermia-induced IL-8 increases the degree of chemoattraction of neutrophils [30]. Increased IL-6 and IL-10 levels have also been reported above 41 °C. However, these factors could lower the presence of IL-2 and IFN-γ, although evidence indicates that their status is unchanged [35]. Temperatures exceeding 41–42 °C shift the cytokine environment from proinflammatory to anti-inflammatory [16]. HSP mediates nitric oxide release via APCs (DCs and macrophages), which are capable of protecting cells from TNF-⍺-induced cell death by inducing HSP70 production [17,63].

##### IL-6

IL-6 is classically known as a protumorigenic cytokine that promotes tumor progression and metastasis [64]. However, there is a line of evidence concerning the antitumor effects of IL-6, which is produced by stromal cells and plays a key role in the hyperthermia-mediated increased trafficking of lymphocytes, such as CD8+ CTLs, across the tumor vasculature [20,38,57,64]. In addition, IL-6 trans-signaling-induced gene transcription increases the expression of (1) acute-phase response elements such as CRP and SAA, (2) chemokines such as CCL21 and CXCL8, and (3) IL-2⍺, which promotes B-cell activation [38]. IL-6 also appears to be essential for the upregulation of E-selectin, P-selectin, L-selectin, and ICAM-1, as shown by IL-6 antibody-mediated blockade in multiple tumor models [20,53,64]. On the other hand, the sIL-6Ra/gp130 complex is upregulated by induced hyperthermia due to increased sIL-6Ra formation and secretion while also increasing gp130 expression in endothelial cells near tumors [20,38,45,57,64]. The second messenger system of IL-6 trans-signaling, STAT3, is also increased by HSP90, which stabilizes STAT3 signaling, resulting in increased gene transcription [38].

This selective nature is supported by the finding that only HEVs and the tumor vasculature are responsive to the IL-6-induced upregulation of adhesion molecules [20]. In the first 5 h after induced hyperthermia, an increase in IL-6 was observed, and these levels returned to normal after 24 h [18].

#### 3.1.6. Immune Cell Trafficking

At 39 °C, induced hyperthermia enhances immune cell trafficking into the tumor microenvironment, improving the antitumor immune responses [15,16,17,18,20,23,29,30,31,36,38,40,45,51,52,53,57,64]. Hyperthermia increases both leukocyte adhesion and diapedesis, increasing the expression of endothelial adhesion molecules such as ICAM-1, L-selectin, E-selectin, P-selectin, and integrins [20,23,30,38,45,53,64]. This makes immune interactions with endothelial cells more stable and resistant to blood flow. Induced hyperthermia also increases blood flow and induces the selective accumulation of lymphocytes in secondary lymphatic organs, facilitating stronger interactions between antigen-presenting cells (APCs) and lymphocytes [20,30,38]. It selectively promotes the transport of naïve and memory T and B lymphocytes, leading to increased tumor infiltration by CD3+, CD4+, and CD8+ cytotoxic T lymphocytes (CTLs) [17,19,23,38,46]. Owing to increased immune cell trafficking, a significant decrease in the number of circulating immune cells (mainly CD8+ CTLs, NK cells, and neutrophils) is attributed to their migration into tissues, not to their death [20,29,38,46,64]. Importantly, induced hyperthermia reduces CD4+ CD25+ FoxP3+ Tregs, shifting the ratio (from 1:5 to 1:35) in favour of CTLs, which aids in the antitumor response [53,64]. Additionally, induced hyperthermia downregulates immunotolerance mechanisms and inhibits genes linked to Treg recruitment and immune suppression. However, some studies suggest that induced hyperthermia might also enhance Treg mobility in certain cases [19].

In summary, induced hyperthermia (~39 °C) exerts a potent immunomodulatory effect by activating both innate and adaptive immune responses. It enhances lymphocyte survival, promotes tumor immune infiltration, and correlates positively with improved survival outcomes in cancer patients. Heat shock proteins (HSPs), particularly HSP70 and HSP90, act as key danger-associated molecular patterns (DAMPs), which initiate and amplify immune activation through TLR2/TLR4-mediated dendritic cell maturation, cytokine release, and efficient antigen presentation. Hyperthermia also augments cytotoxic lymphocyte (CTL) and NK cell activity—facilitated by NKG2D receptor clustering and HSP70 interaction—while boosting IL-2 production, T-cell proliferation, and tumor-directed cytotoxicity. Moreover, macrophage and B-cell activation, together with enhanced neutrophil infiltration, contribute to antitumor immune response and strengthened immune surveillance.

However, these effects are highly temperature-dependent: while immune activation predominates at 39–41 °C, temperatures above 42 °C induce apoptosis of immune cells, anti-inflammatory cytokine dominance, and thermotolerance, thereby diminishing therapeutic efficacy. The dual role of cytokines such as IL-6 further highlights the fine balance between pro- and antitumor effects under thermal stress. Overall, induced hyperthermia reshapes the tumor immune landscape by promoting leukocyte adhesion, improving vascular perfusion, and favouring cytotoxic over regulatory T-cell infiltration. These findings underscore the potential of precisely controlled hyperthermia to synergize with immunotherapies and enhance antitumor immune responses through coordinated activation of multiple immune pathways.

### 3.2. Genome Instability and Mutation

#### Induced Hyperthermia Disrupts the DNA Repair Mechanisms

Defects in the genome maintenance pathways involve issues in detecting DNA damage, activating repair processes, directly repairing damaged DNA, or neutralizing mutagenic molecules before they damage the DNA [65,66]. Above 40 °C, hyperthermia induces replication fork halting by DNA polymerase ⍺ and β inhibition above 40 °C [22,35,45,56,67,68,69,70,71,72,73,74]. However, HSPs are capable of restoring DNA polymerase function and thereby support thermotolerance [47]. In addition, HT causes nuclear proteins to bind to the nuclear matrix, disrupting the proper function of the nuclear matrix in regulating DNA repair, transcription, and DNA supercoiling [27,35,56,71,73]. Induced hyperthermia primarily affects cells in the S phase, leading to chromosomal aberrations [68,75]. Above 40 °C, it also inhibits ATR/Chk1 phosphorylation and DNA polymerase α, halting replication and exposing single-stranded DNA, which can result in single-strand breaks (SSBs), double-strand breaks (DSBs), and cell death. This triggers a mitotic crisis at 43 °C [22,23,35,45,52,67,68,69,70,71,73,75,76]. Induced hyperthermia also affects DNA damage response (DDR) pathways by reducing 53BP1 levels and delaying the assembly of the p-H2AX/MDC1/53BP1 complex, which impairs DNA repair [23,69,75]. At 40 °C, the MRN complex moves to the cytoplasm, but ATM activation still occurs, likely because reactive oxygen species (ROS) break disulfide bonds in ATM dimers. ATM activation leads to cell cycle arrest and delayed DDR activation, particularly causing a pause in nonhomologous end joining (NHEJ) and the inhibition of homologous recombination (HR), resulting in disrupted and impaired DSB repair (Figure 5) [34,69,75,77]. Activation of ATM and ATR increases cell cycle arrest and apoptosis. Direct DNA damage from induced hyperthermia below 42.5 °C is unlikely [16,30,31,45,64,67,68,71,75,76]. This occurs due to damage to nuclear proteins, repair enzymes, and DNA anchoring proteins. The nuclear proteins migrate and aggregate, further disrupting repair [67,71]. HSP70 can partially protect these pathways by restoring protein function and enhancing endonuclease activity by chaperoning APE1, explaining partial thermoresistance [78]. In hypoxic cells, induced hyperthermia causes even greater DDR inhibition than radiotherapy by increasing cytotoxicity, suggesting a synergistic effect of induced hyperthermia and radiotherapy [79]. Additionally, induced hyperthermia affects the homologous recombination (HR) pathway, which is energy expensive, has a slow but precise DDR, and is active during the S- and G-phases [34,68,80,81]. It inhibits key components such as MRE11 by forcing this component to leave the nucleus, preventing the formation of the MRN complex and disrupting HR functionality (Figure 5) [23,54,68,71,75]. In addition, Rad54, XRCC, and BRCA2 are impacted above 40 °C [34,68,80,81]. This leads to an inability to replace RPA foci with Rad51 foci, halting HR initiation [15,23,34,68,75,76,81,82,83]. Induced hyperthermia also changes the conformation of the RPA complex, independent of FancD2 binding [69]. It also alters ATR and Chk1 phosphorylation, resulting in G2/M cell cycle arrest and G1 arrest due to WAF1 expression [34,69,73,75]. While these effects are temporary and functions can recover within hours, the inhibition of HR pathways can increase immune cell infiltration at tumor sites [34]. Furthermore, induced hyperthermia negatively impacts the base excision repair pathway by inhibiting DNA polymerase β and DNA glycosylases, increasing the vulnerability of DNA strands to damage [15,27,35,69,75]. Induced hyperthermia can lead to genomic instability in cells, possibly due to epigenetic changes, but it does not directly cause oncogenesis. However, induced hyperthermia can increase the effects of environmental carcinogens, making it important for patients to avoid exposure to substances such as smoking or alcohol before and after treatment. A study revealed that waiting 24 h between hyperthermia treatment and carcinogen exposure did not increase carcinogenesis [67].

In summary, defects in genomic maintenance compromise the detection and repair of DNA damage, and induced hyperthermia intensifies these vulnerabilities in a temperature-dependent manner. At temperatures exceeding 40 °C, induced hyperthermia disrupts DNA polymerase function—particularly in S-phase cells—resulting in replication fork stalling, chromosomal aberrations, and impaired activation of DNA repair pathways. These effects collectively promote cell cycle arrest and apoptosis, thereby enhancing the cytotoxic efficacy of chemoradiotherapy. Moreover, induced hyperthermia destabilizes homologous recombination by inhibiting essential repair proteins such as MRE11 and BRCA2, leading to G2/M phase arrest and persistent genomic instability. Base excision repair is similarly compromised, further sensitising tumor cells to genotoxic stress. Overall, induced hyperthermia acts as a powerful modulator of DNA repair fidelity, amplifying therapeutic responses while exposing temperature-dependent vulnerabilities in tumor genome stability.

### 3.3. Resisting Cell Death

#### Induced Hyperthermia Modulates Cell Death Pathways

Apoptosis, as part of the programmed cell death process, functions as a barrier to cancer development [84,85,86]. In addition to the loss of TP53, tumors may attain an increased expression profile of antiapoptotic proteins (Bcl-2, Bcl-xL) or survival signals (Igfl/2) by downregulating proapoptotic factors (Bax, Bim, and Puma) or by short-circuiting the extrinsic ligand-induced program [84,87]. Autophagy, induced by Beclin-1, functions at low basal levels in cells but is enhanced by cellular stress, such as nutrient deficiency [88,89]. Certain cytotoxic drugs and radiotherapy can induce elevated levels of autophagy, causing cytoprotection and destroying the killing effect of these stress inducers [90,91]. The proinflammatory and tumor-promoting potential of necrosis could be explained by the explosion of necrotic cells and the release of their contents into the local tissue microenvironment. As a consequence, the released necrotic contents can trigger inflammatory cells in the immune system, resulting in debris removal. However, immune inflammatory cells are capable of tumor promotion (IL-1a release) by enhancing cell proliferation and angiogenesis [90,92]. The survival of cells exposed to induced hyperthermia shows a dose-dependent relationship between the thermal dose and cytotoxicity up to 42–43 °C (the breakpoint). Higher temperatures are associated with an exponential increase in toxicity in healthy cells [35]. Induced hyperthermia stimulates apoptotic and necrotic cell death, and this process is related to the temperature and duration; above 43 °C, evidence shows exponentially more necrosis. However, necrosis also occurs at lower temperatures in combination with radio/chemotherapy [15,16,22,24,25,26,27,30,35,45,48,56,69,70,82,93,94,95,96,97,98,99,100,101]. In the context of apoptosis, induced hyperthermia causes stress-induced apoptosis-like cell death (SICD) in tumor cells via an increase in mitochondrial ROS [93]. Another mechanism involves wtp53 signal activation, resulting in cell cycle arrest, genome stabilization, or the activation of apoptotic signals such as Bax. The activation occurs through p53 phosphorylation via PI3-K family enzymes (PKC, ATM, and ATR), increasing wtp53 activity [96]. Mutated p53 (mp53) is capable of repressing wtp53, and higher levels of mp53 imply lower levels of wtp53 [34,96]. In this respect, induced hyperthermia can help wtp53+ malignant cells regain proliferation control and tumor suppression ability. This finding implies that the status of p53 is indicative of the effectiveness of HT cytotoxicity [33,96,99]. HSP90 can block wtp53 from inducing apoptosis, whereas HSP70 is able to chaperone mp53 and decrease its repressive effect on wtp53 expression [33,45]. Induced hyperthermia decreases miRNA-23a levels and upregulates miRNA-34a, but both miRNAs induce p53-mediated apoptosis [82,97]. Induced hyperthermia reduces Bcl-2 levels and increases Bax levels, resulting in changes in the Bax/Bcl-2 ratio and the induction of apoptosis [23,70,93,95,100]. However, HSP70 can suppress the activation of Bax indirectly via c-Jun N-terminal kinase (JNK) inhibition. JNK acts in a proapoptotic manner through the activation of multiple BH3 proteins, including Bid, Bim, and Bmf, which trigger Bax to cause mitochondrial outer membrane permeabilization (MOMP) [48,100]. In addition, the proapoptotic effects of Bax can be inhibited through the HSP clusterin (sCLU) [47]. Another proposed hyperthermia-mediated SICD pathway involves the activation of caspase-2 to form a complex with its adaptor protein RAIDD, thereby activating caspase-2 and cleaving Bid into t-Bid. In response, t-Bid can stimulate MOMP, causing the release of cytochrome c, resulting in the formation of apoptosomes composed of Apaf-1 and caspase-9 [54,73,100]. Some evidence has identified caspase-9 as a major contributor to the apoptotic fraction, whereas other evidence suggests that this claim is overstated [15,100]. In addition, Bax and/or Bak are activated by t-Bid due to heat-induced conformational changes and by the release of Bim due to cytoskeleton damage [73]. Changes in the cytoskeleton network result in abnormal cell shapes, anoikis, and apoptosis [35,70]. Moreover, induced hyperthermia is a strong activator of JNK, which is capable of phosphorylating Bim and inducing apoptosis [82,100]. Apoptosis-inducing factor (AIF), Smax/Diablo, and other non-caspase pro-apoptotic factors are also activated by induced hyperthermia. In addition, extrinsic death pathways are enhanced by induced hyperthermia, which potentiates SICD signals via Fas/FasL, TRAIL (TNF-related apoptosis-inducing ligand), and TNF-⍺ [54]. As a result of TNF signaling, HSF-1 is downregulated, thus lowering the levels of HSPs in cells. In HSP-dependent tumor cells, this can result in thermosensitization as well as radio- and chemosensitization [100]. At 42 °C and 44 °C, Fas signaling activity is elevated [73]. Increased activation of caspase-3, -8, and -9 and the release of cytochrome c from mitochondria were observed during hyperthermia-induced TRAIL signaling [73]. In addition, FLIP, which inhibits caspase-8 activation, is downregulated. This results in a pro-apoptotic stimulus [73,100]. Induced hyperthermia also causes endoplasmic reticulum stress due to an accumulation of unfolded proteins in the ER, which triggers cell death through Ca^2+^ influx [54,73]. Induced hyperthermia can also activate survival pathways. Induced thermotolerance is associated with the activation of antiapoptotic pathways (Akt, p38, and ERK) [53,93]. At 41.5 °C, necrosis is the most common method of cell death. This effect is enhanced in combination with radiotherapy [22]. Necrotic cells are proinflammatory and activators of antitumor immunity since their membrane bursts, exposing all their contents (tumor antigens and DAMPs) [19,22,23,24,29,30,31,48]. Induced hyperthermia induces protein aggregation via the use of MEKK1 as a signaling sensor. When protein aggregates are too large to be degraded by the proteasome or chaperoned by HSPs, autophagy is required for their degradation [102]. Additionally, hyperthermia-induced macroautophagy is achieved through the activation of small HSPs (HSP-B family), the Bcl2 family proteins Beclin 1 and p53 [102,103]. HSF-1 increases BAG3 expression, and the HSPB8-BAG3 (an HSP70 cochaperone) complex marks aggregated proteins for autophagy [82,102]. The HSP70-BAG3 complex increases NF-kB, HIF-1⍺, and p21, leading to antiapoptotic, prosurvival effects [82]. On the other hand, HSP70 and 90 induce a special form of autophagy called chaperone-mediated autophagy [82,102]. Induced hyperthermia significantly increases the levels of phosphorylated AMPK while decreasing the levels of phosphorylated mTOR, inducing autophagy via damage to ER protein aggregation [103]. In senescent tumor cells, autophagy can facilitate their survival through many stressors [104]. In response to induced hyperthermia, cells in the M and S phases undergo slow and controlled apoptotic cell death, while cells in the G1 phase are relatively hyperthermia resistant but eventually undergo necrotic cell death at a certain thermal dose [16,27,35,56,70,93]. Cells in the S phase are the most sensitive to induced hyperthermia; they are characterized by visible damage to their mitotic apparatus, causing failure to move into the M phase and the formation of polyploid nonclonogenic cells [16,27,35,56,70,82,93]. Induced hyperthermia stress-induced apoptosis-like cell death (SIa1CD) activates dendritic cells, followed by increased IL-12 expression [18,53,105]. The immunogenicity of apoptosis depends on the trigger, while homeostatic apoptotic cells are anti-inflammatory, stressed tumor cells are proinflammatory [18,31].

In summary, apoptosis represents a fundamental barrier to cancer progression, frequently compromised by TP53 mutations and dysregulated apoptotic protein expression. Induced hyperthermia restores this barrier through temperature- and duration-dependent induction of cell death via apoptosis and necrosis. The apoptotic response is modulated by TP53 status and miRNA regulation, which together influence key signaling cascades. Induced hyperthermia downregulates the antiapoptotic protein Bcl-2 while upregulating Bax, shifting the Bax/Bcl-2 ratio toward programmed cell death. Concurrently, stress-activated kinases such as JNK, cytoskeletal disruption, and caspase-2 activation amplify intrinsic apoptotic signaling, while extrinsic mechanisms—mediated by Fas/FasL and TNF-α—further enhance cell death and endoplasmic reticulum stress responses.

Protective mechanisms, including HSP70 and secretory clusterin (sCLU), counterbalance these effects by promoting thermotolerance and suppressing apoptosis, underscoring the dual nature of hyperthermia-induced stress. At temperatures around 41.5 °C, necrotic pathways predominate, releasing danger signals that fuel inflammation and antitumor immunity. Simultaneously, hyperthermia-induced protein aggregation triggers autophagy for cytoprotection and degradation of damaged proteins, an effect tightly regulated by HSPs. Moreover, induced hyperthermia selectively induces apoptosis in specific cell cycle phases while promoting proinflammatory cytokine release, including IL-12, thereby linking cellular stress responses to enhanced immune activation.

Collectively, these mechanisms position induced hyperthermia as a powerful modulator of cells, capable of tipping the balance between survival and death to favour tumor regression and immune stimulation when optimally applied.

### 3.4. Deregulating Cellular Energetics

#### Induced Hyperthermia Disrupts Cancer Cell Metabolism

Uncontrolled cancer cell proliferation requires adjustment of energy metabolism to fuel cell growth and division. The adjustment starts with the upregulation of glucose transporters, which increases glucose uptake [106]. However, tumor cells have relatively poor efficiency in generating ATP via glycolysis. Considering the two subpopulations of cancer cells that differ in their energy-generating pathways, one consists of glucose-dependent (‘Warburg effect’) cells that secrete lactate, whereas the other imports and utilizes the produced lactate as its main energy source, employing part of the citric acid cycle [107,108]. Additionally, oxygenation, ranging from normoxia to hypoxia, is not necessarily static in tumors but instead fluctuates temporally and regionally, likely as a result of the instability and chaotic organization of the tumor-associated neovasculature [109]. Induced hyperthermia acts as a sensitizer by improving tumor oxygenation [110]. In addition, it has significant effects on cellular metabolism, leading to reduced ATP production and increased enzyme activities [15,62,111,112]. This shift in metabolism results in greater purine breakdown, generating protons and reactive oxygen species (ROS) in tumor cells [113]. Mitochondrial oxidative phosphorylation is inhibited by Ca^2+^ influx, NO, and ROS, particularly in dendritic cells [59,112]. This effect also remains permanent even after the patient returns to normothermia [59]. Induced hyperthermia also promotes glucose utilization through the upregulation of HIF-1, leading to increased glycolytic enzymes and lactic acid production [57]. A switch to anaerobic metabolism occurs in response to the upregulation of PDK-1 [52,57]. While tumor ATP and glucose levels remain unchanged after exposure to hyperthermia, lactate levels increase up to 1.5-fold, contributing to greater tumor aggressiveness and metastatic progression at 42–43 °C [57]. Above 42 °C, cancer cells experience cytotoxic damage due to increased production of ROS, marked by a depletion of reduced glutathione and an increase in oxidized glutathione [57,62,73,75,82,93,114,115,116]. This increase in ROS is also linked to the conversion of xanthine dehydrogenase to its oxidase form, the induction of energy enzymes, increased oxygen delivery, alterations in mitochondrial metabolism, and increased NADPH oxidase activity [53,57]. The production of mitochondrial superoxide, alongside the contributions of mitochondrial complexes I and II, disrupts mitochondrial functions, causing impaired lipid and protein synthesis [62,73,82]. ROS can inflict damage to mitochondrial DNA, which is less repairable than damage to nuclear DNA [117,118,119]. Induced hyperthermia reduces the levels and activity of superoxide dismutase 1 (SOD-1) while increasing manganese superoxide dismutase (MnSOD) levels and lowering the levels of peroxiredoxin 2, indicating an antioxidant response [62]. All these effects result in elevated ROS levels, which can then provoke genomic instability and inhibit cell proliferation by blocking cell cycle progression in the G1, S, or G2 phase and can even induce apoptosis [114]. ROS also displace complexes involving Beclin-, Bcl-2- and Bcl-xL/McI-1, leading to increased autophagy [82]. Additionally, hyperthermia-induced ROS increase the expression of proinflammatory genes such as TNF-α, NF-kB, and JNK [62]. While low ROS levels can promote tumor growth, excessive ROS can overwhelm cellular defenses, resulting in cell death [57]. ROS influence various signaling pathways (epidermal growth factor receptor, c-Src, p38 mitogen-activated protein kinase, extracellular-signal related kinases-1 and -2, Ras/Raf, stress-activated protein kinase, Jun N-terminal kinase, and Akt/protein kinase B), potentially altering cellular physiology through “sulfhydryl switches” [116]. Finally, induced hyperthermia inhibits the pentose phosphate pathway (PPP), which may reduce antioxidant responses to ROS and affect signaling related to radio- and chemosensitization, with HSP27 modulating ROS levels through the glutathione pathway [116]. The resistance of tumor cells to induced hyperthermia at a constant acidic pH is due to elevated levels of HSPs at 42 °C. However, depending on the tumor type, resistance can transition to sensitization beyond a specific acidity threshold. For example, melanoma shows sensitization at pH 6.7 and lower, possibly due to its ability to excrete H+ ions to maintain the intracellular pH. Tumor cells acutely exposed to acidic conditions are more likely to be sensitive to hyperthermia at 42 °C [27,32,55,57,111,112,120].

In summary, the uncontrolled proliferation of cancer cells drives profound metabolic reprogramming to sustain their high energy demands. Induced hyperthermia disrupts this metabolic adaptability by decreasing ATP production, accelerating enzymatic activity, and promoting purine catabolism, which collectively elevate reactive oxygen species (ROS) levels. This oxidative stress compromises mitochondrial integrity and induces cytotoxic damage, particularly at temperatures exceeding 42 °C. Concurrently, induced hyperthermia increases lactate accumulation, a hallmark of metabolic stress that can enhance tumor aggressiveness and alter the tumor microenvironment.

Beyond metabolic disruption, induced hyperthermia upregulates proinflammatory gene expression and modulates key signaling pathways involved in cell survival and stress responses. Interestingly, tumor cell resistance to induced hyperthermia under acidic conditions can shift toward sensitivity depending on tumor type and microenvironmental pH, highlighting the intricate interplay between heat, metabolism, and cellular context.

Collectively, these findings position induced hyperthermia as a potent modulator of tumor metabolism—capable of undermining cancer’s energetic resilience while sensitizing malignant cells to oxidative and inflammatory stress, thereby amplifying therapeutic efficacy.

### 3.5. Inducing Angiogenesis

#### Induced Hyperthermia Exerts Dual Effects on Tumor Angiogenesis

The sustainability of tumor angiogenesis is regulated by the VEGF-⍺ gene and proangiogenic signals such as fibroblast growth factor or TSP-1 (trombospondin-1) suppression [121]. Neutrophils, macrophages, and mast cells can infiltrate progressive tumors and sustain angiogenesis [122]. Conflicting evidence exists concerning the effect of induced hyperthermia on angiogenesis. On the one hand, it has an antiangiogenic effect by inhibiting endothelial cell proliferation in the tumor vasculature, primarily through HIF-1 and VEGF inhibition, with PAI-1 upregulation also playing a role [57,112]. On the other hand, induced hyperthermia stimulates angiogenesis by increasing VEGF and angiopoietin expression, especially at 42 °C, through HIF-1 and VEGF upregulation, ROS production, and HSP90 chaperoning [41,52,53,57]. The balance between proangiogenic (VEGF) and antiangiogenic (PAI-1) factors, both of which are regulated by HIF-1, explains the mixed reports on their effects [106,107]. Hypoxia is a key factor in initiating tumor angiogenesis, including the role of oncogenes and the loss of tumor suppressor genes [117]. Compared with radio/chemotherapy- and nutrient-deprived cells, tumor cells in hypoxic, acidic, and nutrient-deprived states are more susceptible to hyperthermic cytotoxicity [27,31,41,42,43,52,54,55,56,58,79,93,111,112,117,118,120,123,124,125]. The increase in thermosensitivity is caused by ATP depletion from increased metabolic activity [15,101,111,126]. Hyperthermia below 42 °C increases blood flow, oxygenation, vascular permeability, and drug delivery, reducing interstitial pressure in tumors [15,22,27,31,35,45,51,52,53,54,55,57,58,64,79,101,111,112,118,120,123,124,127,128]. This leads to a transient increase in oxygen levels in the tumor microenvironment [15,22,31,35,42,45,52,53,54,56,57,58,79,112,120,123,124,127,128]. In general, tumor hypoxia (pO_2_ of less than 2.5–10 mmHg) is linked to more aggressive cancer, treatment resistance, and poor patient outcomes [58,129]. Because of induced hyperthermia, oxygenation is greater than with radio/chemotherapy alone [112,127]. The level of oxygenation in tumors following induced hyperthermia treatment varies depending on the location of the tumor [58,128]. Hyperoxygenation can last from a few minutes to 24–48 h after treatment at temperatures above 43 °C [53,55,111,112,118,124,128]. Increased expression of iNOS post-treatment may contribute to a peak in oxygen levels [58,112,130]. Induced hyperthermia treatment can lead to a significant reduction in tumor cells without permanently collapsing the tumor vasculature, resulting in an excess of oxygen in the tumor tissue [22,112,118,124,128]. At 41–42 °C, there is an increase in oxygen consumption by cells due to enzyme activity and low direct cytotoxicity, whereas higher temperatures (>42 °C) can decrease the oxygen content in the tumor mass [112,118]. After increased blood flow, the tumor vasculature collapses, triggered by thrombosis, which can lead to ischaemic necrosis, particularly in tumor tissues with irregular vascular organization [24,31,34,35,45,58,112,120,124,128,130]. Induced hyperthermia treatment at temperatures greater than 42 °C activates blood coagulation and affects RBC membrane viscosity, reducing the oxygen-carrying capacity and leading to hypoxic, acidic, and necrotic tumor tissue [35,112]. The tissue selectivity of induced hyperthermia has been described as ‘the steal effect’. This effect causes healthy tissue to thermoregulate via vasodilatation, causing an increase in blood flow of up to 15-fold, whereas tumors fail to regulate heat effectively. Tumor cytotoxicity is greater in tumors with poor vascularization, but well-vascularized tumors may be difficult to heat. This highlights the importance of tumor vascularization in induced hyperthermia treatment efficacy [35,56,112,117,118,120,124].

In summary, tumor angiogenesis is orchestrated by complex interactions between proangiogenic mediators such as VEGF-α, HIF-1, and immune cell-derived cytokines from neutrophils and macrophages. Induced hyperthermia exerts a dual and temperature-dependent influence on this process: at moderate levels, it suppresses endothelial cell proliferation by downregulating HIF-1 and VEGF signaling, thereby restricting neovascularization; however, at higher temperatures, it can paradoxically enhance VEGF expression, promoting angiogenic activity.

Tumor hypoxia, a critical driver of malignancy and poor clinical outcomes, is transiently alleviated by induced hyperthermia through improved perfusion and oxygen delivery. Nonetheless, this benefit may be counterbalanced by subsequent vascular damage and coagulation effects at elevated temperatures. The observed “steal effect” further underscores the pivotal role of tumor vascularization in determining therapeutic response and treatment efficacy.

Collectively, these findings highlight induced hyperthermia’s complex, bidirectional modulation of angiogenesis—simultaneously capable of improving tumor oxygenation and sensitising tumors to therapy, while also carrying the potential to foster angiogenic rebound under specific thermal and microenvironmental conditions.

### 3.6. Activating Invasion and Metastasis

#### Induced Hyperthermia Has a Dual Role in the Process of Cancer Metastasis, Addressing the Protumoral Effect

HSF-1, which is increased and released by induced hyperthermia, is capable of inducing metastasis by silencing antimetastatic genes and promoting promodulatory signaling cascades. This response is used by transformed cells to increase their degree of malignancy, including elevated levels of HSPs [131]. HSF-1 is also capable of promoting the expression of MTA-1, a corepressor of oestrogen-induced antimetastatic genes, indicating the potential of induced hyperthermia to promote metastasis [131]. In addition, HSF-1 can also override cell cycle checkpoints, resulting in proliferation, aneuploidy, and enhanced metastasis in tumor cells [48]. Another type of activation involves HSP-B1 as a regulator of apoptosis and metastasis, promoting both processes when it is present at relatively high levels [49]. A higher level of HIF-1 also leads to increased EMT progression [58]. Induced hyperthermia increases the fluidity and permeability of the cellular membrane, resulting in induced changes in transmembrane transport, increased drug uptake, and the induction of immune cell activation [15,16,35,45,51,52,53,55,56,70,71,73,93,119]. The changes in transmembrane transport lead to increased Ca^2+^ and Na^+^ influx, K^+^ efflux, and an undefined change in Mg^2+^ [35,45,70,93]. These changes are not correlated with cytotoxicity [45,93]. However, some authors claim that hyperthermia-induced Ca^2+^ influx plays a proapoptotic role [73,93]. Induced hyperthermia leads to an increased presence of membrane phospholipid precursors, signifying increased breakdown or decreased synthesis of phospholipids [111,112]. Research has proposed that plasma membrane changes could be involved in hyperthermia ‘sensing’ and the activation of HSPs as protective factors mediated by the mGFP-GPI homoassociation [119]. Another factor involved in hyperthermia-induced plasma membrane signaling is changes in the lipid raft composition caused by the creation of Cer-rafts via the substitution of choline with Cer. These rafts activate the PI3-kinase, Akt, and glycogen kinase 3 pathways, transmitting stress signals to the cell [119]. Hyperthermia-induced changes in the mitochondrial membrane potential cause redox changes in cells, resulting in disruption and dysfunction of nuclear proteins. These cells are then more vulnerable to further induced cytotoxicity [27,35,71,73]. HSPB11 is believed to be capable of preventing H_2_O_2_- and taxol-induced cell death by stabilising the mitochondrial membrane and the plasma membrane, inducing HSP90 and activating PI-3-Kinase-Akt signaling [119]. Induced hyperthermia leads to protein damage by unfolding, denaturation, and aggregation, causing loss of function in the mitochondria and lysosomes, swelling of the nucleoli, and protein complex deposition. All of these effects contribute to the result of mitotic crisis [15,22,27,35,43,45,56,73,74,82,93]. In addition, protein synthesis is inhibited in a temperature-dependent manner (42–45 °C), especially DNA synthesis, which remains inhibited for a longer period of time [27,35,55,56]. The breakpoint of direct cytotoxicity relies on cytosolic and membrane protein denaturation, starting at 40 °C, with changes in the cytoskeleton, repair enzymes, and membranes [43,56,82]. Considering the observed cytoskeletal changes, induced hyperthermia decreases the levels of soluble actin and tubulin while increasing the level of phosphorylated cofilin 2, indicating a loss of microtubule structures while increasing the number of stable actin microfilaments. This can lead to a breakdown of the intracellular transport system [62,73]. In addition, induced hyperthermia causes rearrangement and/or disassembly and spectrin aggregation of the cytoskeleton [16,55,73,74,82,93].

In summary, hyperthermia-induced activation of heat shock factor 1 (HSF-1) exerts a dual influence on tumor biology, simultaneously promoting malignant transformation and therapeutic sensitization. Elevated HSF-1 levels drive metastasis by suppressing antimetastatic gene expression and activating pro-metastatic signaling pathways, including the upregulation of metastasis-associated protein 1 (MTA-1), which interferes with oestrogen-mediated tumor suppression. This transcriptional reprogramming enhances cellular malignancy, genomic instability, and unchecked proliferation through the bypassing of cell cycle checkpoints.

Conversely, induced hyperthermia disrupts membrane integrity and cytoskeletal stability, increasing permeability, drug uptake, and immune recognition, while inducing mitochondrial dysfunction and proteotoxic stress. Together, these effects underscore the paradoxical role of HSF-1 activation and thermal stress—capable of driving both tumor progression and therapeutic vulnerability, depending on temperature intensity, duration, and cellular context.

### 3.7. Enabling Replicative Immortality

#### Induced Hyperthermia Supports Replicative Immortality of Cancer Cells

Telomeres, which protect the ends of chromosomes, are involved in unlimited proliferation via upregulated expression of telomerase [132,133]. Telomerase activity is correlated with resistance to the induction of both senescence and apoptosis [134,135,136]. During malignant transformation, telomerase activation is a prerequisite for the ability of malignant cells to proliferate ad infinitum. HSP23 and HSP90 are essential for preventing hTERT ubiquitination and proteasomal degradation. In the presence of HSP70, induced by HT, the assembly of hTERT is essentially improved [78]. HSP70-deficient cells display rapid telomere depletion and dysfunction, resulting in chromosomal end-end associations and mitotic crisis. The same cells presented increased radiotherapy cytotoxicity when 3 Gy was combined with hyperthermia at 43 °C for 30 min [78]. Two modes of action are proposed to be important for the activation of telomerase by induced hyperthermia: (1) hyperthermia directly influences telomerase functionality, and (2) hyperthermia changes the target-binding sites on the telomere chromatin or nuclear matrix [77,78].

In summary, telomeres safeguard chromosomal integrity and sustain cellular proliferation through telomerase activation. Induced hyperthermia enhances telomerase stability and activity by promoting HSP-mediated assembly and protection of the hTERT complex, thereby supporting tumor cell survival and replication. This interplay between induced hyperthermia, HSPs, and telomerase highlights a critical thermoregulatory mechanism underpinning cancer persistence and potential therapeutic vulnerability.

## 4. Discussion

This systematic review provides a comprehensive analysis of induced hyperthermia’s (HT) effects on the hallmarks of cancer, revealing that seven of the ten hallmarks are directly influenced by this treatment modality (Figure 6). While prior studies have primarily emphasized hyperthermia’s immunomodulatory and chemosensitizing effects, our findings expand this perspective, showing that HT exerts multi-layered, temperature-dependent, and sometimes dualistic effects across diverse cancer traits. These insights not only deepen the mechanistic understanding but also offer guidance for optimizing clinical applications.

### 4.1. Temperature-Dependent Effects

The most well-documented mechanism of action is the activation of the immune system (39–41 °C), followed by genome instability and mutation (41 °C). Both hallmarks yield compelling evidence elucidating the mechanisms underlying induced hyperthermia’s mode of action.

At mild, fever-range temperatures (39–41 °C), hyperthermia enhances antitumor immunity by boosting lymphocyte survival, activating antigen-presenting cells (APCs), and increasing the cytotoxicity of natural killer (NK) cells and cytotoxic T lymphocytes (CTLs). Heat shock proteins (HSPs), particularly HSP70 and HSP90, act as danger signals (DAMPs) that promote antigen presentation, APC maturation, and effector cell activation. Remarkably, HSP70 and 90 have a central role in different immunological reactions, including the potential of activating physiological antitumor effects. Fever-range hyperthermia also induces transient proinflammatory cytokine release, enhancing immune recruitment and tumor killing. A recent publication also highlights the immune-destructive function of induced hyperthermia in various tumor types [137,138]. However, excessive or repeated HT (>42 °C) leads to immune suppression, apoptosis of immune cells, anti-inflammatory cytokine dominance, and thermotolerance, which can reduce therapeutic benefits [15,16]. These findings caution against the frequent or repetitive application of systemic hyperthermia (>42 °C) in cancer treatment.

The induced hyperthermia confers a mechanistic profile that differs from the classical oncology standard-of-care protocols (chemoradioimmunotherapy), potentially leading to divergent biological and clinical effects.

HT also destabilizes genomic maintenance systems, particularly above 41 °C. It impairs DNA polymerases α and β, halts replication forks, and induces single- and double-strand breaks (SSBs and DSBs). Nuclear matrix disruption further compromises repair pathways like base excision repair. The proteins of homologous recombination are destabilised, leading to cell cycle arrest and increased sensitivity to chemoradiotherapy, even under hypoxia. Consequently, induced hyperthermia functions as a synergistic adjunct to chemotherapy and radiotherapy, enhancing their therapeutic efficacy through complementary biological mechanisms.

The other hallmarks of cancer are also impacted by induced hyperthermia. As described in this systematic review, induced hyperthermia influences tumor cell death pathways, shifting the balance between apoptosis, necrosis, and autophagy based on temperature and duration. Apoptosis dominates at 41–43 °C, mediated mainly by p53 activation. Mutant p53 reduces HT efficacy, while HSP70 can restore pro-apoptotic signaling. Necrosis emerges at >41.5 °C, especially in combination with radiotherapy, releasing DAMPs that stimulate immunity. Autophagy helps degrade damaged proteins. The cell cycle sensitivity varies: S-phase cells are most vulnerable (apoptosis), while G1 cells resist but eventually undergo necrosis. HT disrupts the tumor metabolism by suppressing mitochondrial ATP production and inducing oxidative and glycolytic stress. In addition, it generates excessive ROS via multiple enzymatic and mitochondrial sources, damaging DNA, depleting antioxidants, and triggering apoptosis, autophagy, and inflammatory signaling. The pentose phosphate pathway is inhibited, compounding oxidative stress and enhancing therapy sensitivity. In terms of angiogenesis, as one of the other hallmarks of cancer, induced hyperthermia has temperature-dependent effects on the tumor blood vessels. At temperatures <42 °C, HT improves perfusion, oxygenation, and drug delivery by reducing interstitial pressure and transiently suppressing hypoxia. At temperatures >42 °C, HT triggers vascular collapse via coagulation, RBC changes, and thrombosis, leading to ischemic necrosis. The balance between VEGF-driven angiogenesis and PAI-1-mediated suppression, both under HIF-1 control, determines the net outcomes. HT induces structural and biochemical stress—membrane fluidity, ionic fluxes, cytoskeletal disruption, mitochondrial dysfunction—that can sensitize cells to death or drive adaptive survival. The cytoskeletal remodeling (actin/tubulin disruption) and protein damage contribute to mitotic crisis. Besides the antitumoral effects, HT can promote metastasis via HSF-1 activation and upregulation of HSPs, MTA-1, and EMT (via HIF-1). HT supports tumor proliferation by stabilizing telomerase. HSP23, HSP70, and HSP90 protect and assemble hTERT, preventing its degradation and promoting telomere maintenance. HT may directly enhance telomerase activity via protein stabilization and structural changes in telomeric chromatin. Targeting HSP70 or disrupting telomerase assembly during HT could enhance the cytotoxicity of radiotherapy or chemotherapy. The potential synergies or antagonistic hyperthermia effects on the hallmarks of cancer remain poorly understood, and the role of thermotolerance in this context requires further investigation. Besides the described effects of hyperthermia in the first two hallmarks of cancer, the impact of induced hyperthermia on other hallmarks remains underexplored and requires deeper mechanistic research. While certain data provide strong evidence, clinical translation remains a key challenge.

In summary, our review highlights that the antitumoral versus protumoral consequences of induced hyperthermia are highly temperature- and hallmark-dependent. At fever-range temperatures (39–41 °C), HT predominantly enhances antitumor immunity by promoting lymphocyte survival, activating antigen-presenting cells (APCs), and increasing the cytotoxicity of natural killer (NK) cells and cytotoxic T lymphocytes (CTLs). Heat shock proteins (HSP70, HSP90) act as danger signals (DAMPs), facilitating antigen presentation, APC maturation, and effector cell activation. Mild hyperthermia also induces transient proinflammatory cytokine release, further enhancing immune recruitment and tumor cell killing.

In contrast, excessive or repeated exposure (>42 °C) can lead to immune suppression, apoptosis of immune cells, anti-inflammatory cytokine dominance, and thermotolerance, potentially limiting therapeutic efficacy. These findings underscore the importance of defining temperature-specific therapeutic windows to maximize antitumoral effects while minimizing adaptive or protumoral responses. A summary table outlining the effects of hyperthermia across the various hallmarks of cancer is provided in the Appendix A.

### 4.2. Clinical Translation and Future Directions

Further research in clinical settings is necessary to unravel the relative importance of each hallmark to the others. Based on the current evidence, the most clinically promising mechanism of action is the immunological activation observed at temperatures (39–41 °C). This temperature window is consistently achievable in patients, well-tolerated, and has demonstrated reproducible systemic effects, including increased NK-cell activity, enhanced dendritic cell maturation, favorable cytokine shifts, and upregulation of heat-shock responses. In contrast, higher temperatures (>42 °C) are more difficult to achieve safely in humans at the whole-body level and therefore remain less clinically feasible at present. Thus, current data support 39–41 °C as the optimal therapeutic range for systemic hyperthermia, with immune activation being the most relevant and translatable biological mechanism for clinical application.

To identify the strongest mode of action with the long-term effect and to fully harness induced hyperthermia’s therapeutic potential, well-designed clinical trials are needed to define its optimal application, including target temperature, duration, timing, synergy with standard-of-care treatments, and the effect on healthy cells. To move the field forward, future studies should focus on defining temperature-specific therapeutic windows and identifying predictive and prognostic biomarkers (e.g., HSP expression, DDR pathway activation, or immune infiltration profiles) that can guide patient selection and treatment planning. In addition, well-designed clinical trials should incorporate stratification based on tumor type, applied temperature, and exposure duration, alongside real-time biological monitoring to quantify the balance between pro- and antitumoral mechanisms. The integration of induced hyperthermia with immunotherapy or DNA repair inhibitors may represent a novel promising path toward durable responses. Such approaches will allow for more precise, patient-tailored hyperthermia protocols and may help maximize antitumoral effects while minimizing protumoral adaptations.

Whole-body hyperthermia (WBHT) remains the only induced hyperthermia modality available for cancer patients with metastatic disease. Because it induces a systemic heat response, WBHT uniquely affects multiple cancer hallmarks simultaneously, distinguishing it from localized hyperthermia techniques that act only within the tumor microenvironment. To assess the clinical feasibility of WBHT, a first-in-human clinical trial has been completed, exploring its safety and tolerability (ClinicalTrials.gov ID:NCT04467593). In addition, phase II clinical trials of WBHT are currently recruiting patients in randomized controlled clinical trials (ClinicalTrials.gov ID: NCT05821166; ID: NCT06249750).

Nonetheless, the biological systemic effect of WBHT based on the hallmarks of cancer concept requires further investigation to elucidate the underlying mechanisms and associated therapeutic implications.

**Figure 6 cancers-17-03824-f006:**
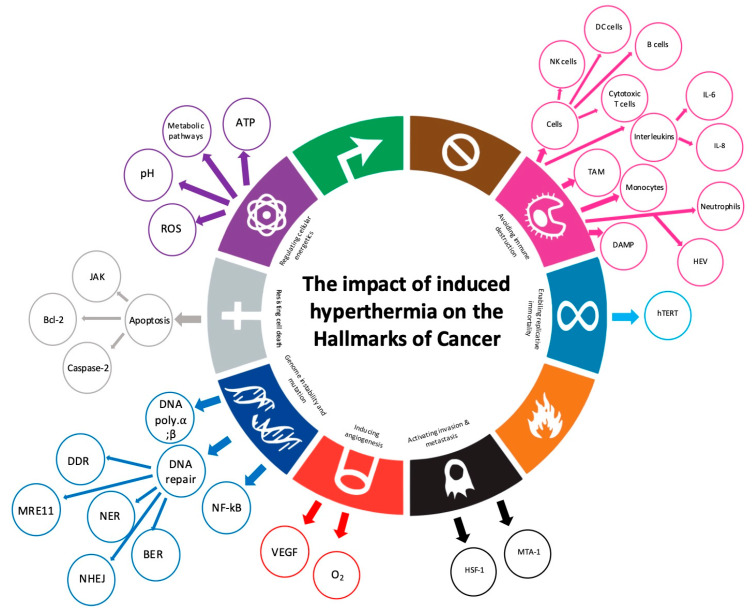
Seven out of the ten hallmarks of cancer are involved in the induced hyperthermia mode of action. Adapted from Rebelo, R et al., 2021 [139].

### 4.3. Critical Discussion of the Reviewed Literature

The body of literature reviewed in this systematic analysis offers several notable strengths while also exhibiting important limitations that shape the interpretation of induced hyperthermia’s effects on the hallmarks of cancer. A major strength of the current evidence is the consistency of findings across multiple independent research groups, particularly regarding immune activation at 39–41 °C and DNA repair inhibition above 41 °C. These mechanisms have been repeatedly validated using diverse methodological approaches—ranging from molecular assays and thermal dose–response studies to animal models—supporting the robustness of key temperature-dependent effects. Additionally, the literature provides a mechanistic basis, with detailed descriptions of HSP biology, cell death pathways, vascular responses, and genomic instability, offering a coherent framework for understanding how thermal stress interacts with fundamental cancer traits. However, despite these strengths, several limitations compromise the overall quality and translational relevance of the current evidence. First, the majority of studies are preclinical, relying predominantly on systems that cannot fully capture the complex thermophysiology of human tissue, systemic heat responses, or tumor microenvironment heterogeneity. This limits the generalizability of mechanistic findings. Second, this review synthesizes evidence primarily from reviews or systematic reviews rather than primary experimental studies, which introduces an additional layer of interpretation and may amplify limitations or biases inherent in the secondary literature. Third, there is considerable methodological heterogeneity across studies, including variations in temperature accuracy, heating methods, exposure duration, and reporting of thermal dose. Such inconsistencies hamper comparability and may underlie conflicting results. Fourth, for several hallmarks—particularly invasion and metastasis, replicative immortality—the available data remain sparse, inconsistent, or of variable quality, with some studies lacking mechanistic depth or adequate controls. Fifth, the literature often examines single hallmarks, overlooking the interconnectedness of cancer traits and the synergistic or antagonistic interplay triggered by thermal stress. Finally, clinical evidence is still limited, with few controlled trials, small sample sizes, and inadequate biomarker integration, restricting conclusions about clinical efficacy or optimal therapeutic windows. Overall, the literature demonstrates strong mechanistic plausibility, substantial cross-study coherence in core pathways, and clear temperature-dependent patterns—representing the main strengths of the evidence. Yet, the variable methodological quality, limited clinical validation, and incomplete hallmark coverage highlight significant gaps. Addressing these weaknesses through standardized hyperthermia protocols, integrative multi-hallmark investigations, and biomarker-driven clinical trials will be essential for translating the promising mechanistic insights of hyperthermia into meaningful therapeutic outcomes.

## 5. Conclusions

This systematic review demonstrates that induced hyperthermia (HT) influences at least seven of the ten hallmarks of cancer, revealing a level of biological complexity far exceeding its traditional roles as an immunomodulatory and chemosensitizing agent. The findings underscore that hyperthermia influences highly temperature-dependent and sometimes dualistic effects, where antitumoral and protumoral mechanisms emerge according to the thermal dose, treatment duration, and tumor context. At temperatures (39–41 °C), hyperthermia predominantly enhances antitumor immunity. Heat shock proteins, especially HSP70 and HSP90, act as DAMPs that facilitate antigen presentation and effector cell activation, supporting tumor rejection. These effects distinguish mild hyperthermia from standard therapies and highlight its potential as an immune-enhancing adjunct. At higher temperatures (>41–42 °C), hyperthermia amplifies genomic, metabolic, and structural stress within tumor cells. It induces DNA damage, impairs repair pathways, disrupts mitochondrial function, enhances ROS production, alters the cytoskeleton, and modulates angiogenesis in a temperature-specific manner. These mechanisms sensitize tumors to radiotherapy and chemotherapy and can result in apoptosis, necrosis, or autophagy depending on treatment conditions. However, excessive or repeated high-temperature hyperthermia can also drive immune suppression, thermotolerance, EMT activation, and telomerase stabilization, emphasizing the importance of precisely controlled thermal dosing. Despite these rich mechanistic insights, major challenges remain for clinical translation. The interplay between hallmarks, the risk of protumoral adaptations, and the systemic effects of whole-body hyperthermia (WBHT) requires deeper investigation. Early clinical trials have established WBHT’s feasibility, but optimal therapeutic windows, biomarker-guided patient selection, and rational combination strategies remain to be defined.

Overall, hyperthermia is a uniquely multi-targeted oncology modality capable of modulating diverse cancer hallmarks, but its therapeutic value depends on rigorous control of temperature, timing, and patient-specific biology. Future trials incorporating temperature stratification, real-time biological monitoring, and integration with immunotherapy or DNA damage–response inhibitors may enable precise, patient-tailored hyperthermia protocols that maximize antitumoral effects while minimizing protumoral adaptations.

## Figures and Tables

**Figure 1 cancers-17-03824-f001:**
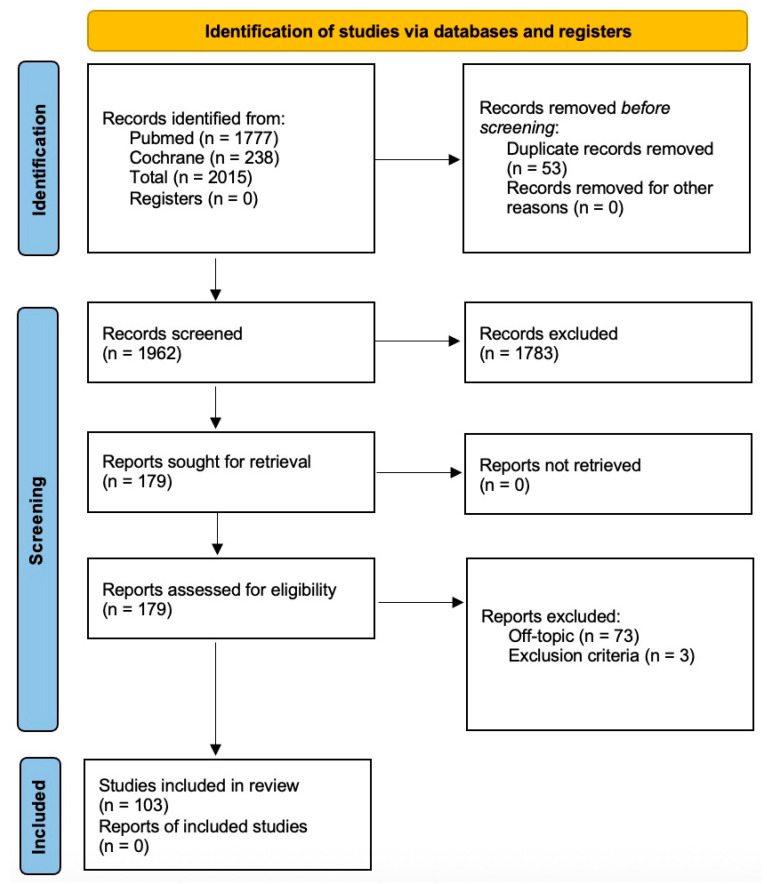
Flow diagram illustrating the identification, screening, and selection of the eligible articles.

**Figure 2 cancers-17-03824-f002:**
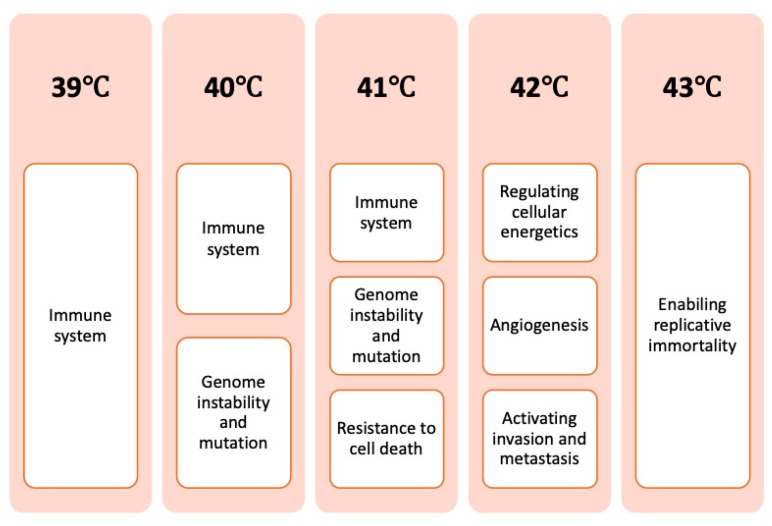
Overview of how the hallmarks of cancer are affected by different temperature intervals.

**Figure 3 cancers-17-03824-f003:**
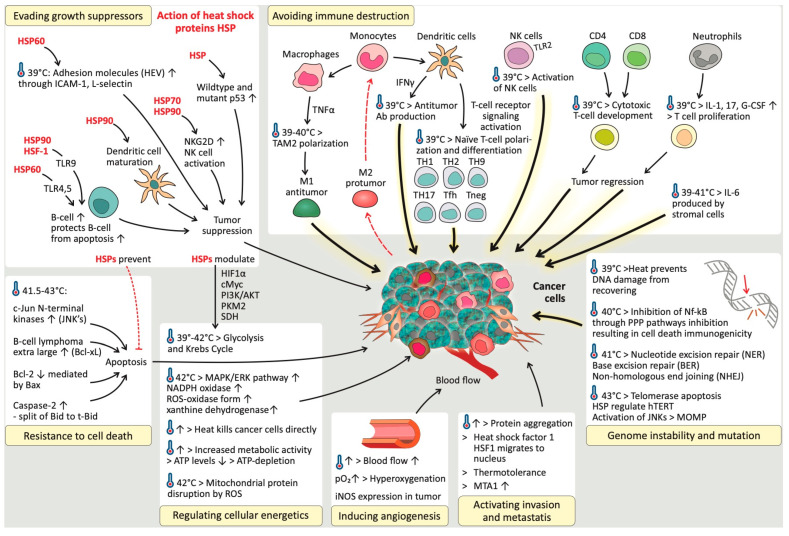
The impact of induced hyperthermia on the hallmarks of cancer.

**Figure 4 cancers-17-03824-f004:**
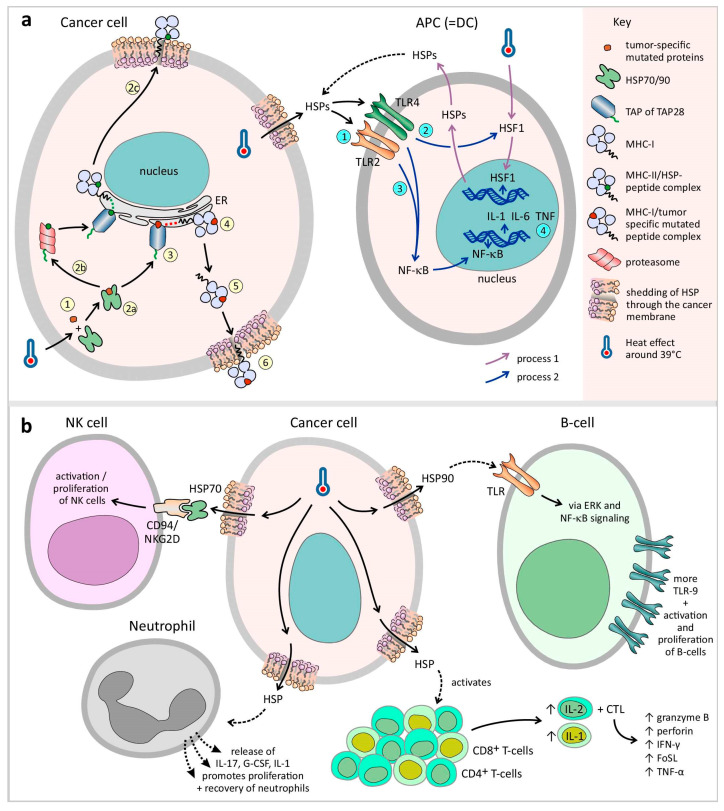
Induced hyperthermia triggers immune-mediated cancer destruction: (**a**) Cancer cell: induced hyperthermia upregulates the HSPs. The role of HSPs as specialized carriers focuses on presenting chaperoned proteins to immune cells, including tumor-specific mutated proteins that are not recognized as ‘self’ epitopes (step 1). In detail, HSP70 and 90 transport the epitope to the endoplasmic reticulum (ER) through a transporter associated with antigen presentation (TAP) (step 2a–3). In the ER, the epitope binds to gp96, which in turn transfers the epitopes to the MHC-I complexes (steps 4–6). In addition, HSP chaperones are degraded by proteasomes and TAP28, which results in peptides that can be presented to MHC-II (step 2b–2c). APC (=DC): The activation of HSPs by TLR2/TLR4 causes the activation of HSF1 signaling followed by NF-kB signaling in DCs, resulting in the release of cytokines (IL-1, IL-6, and TNF) and tumor antigens, which can be processed by APCs (steps 1–4). (**b**) As described above, induced hyperthermia upregulates the HSPs, and HSP70 increases the activity of NK cells by binding to the CD94/NKG2D receptor superfamily, resulting in their activation and proliferation. The proliferation and recovery of neutrophils are promoted by the release of IL-17, G-CSF, and IL-1. HSP-activated B cells express more TLR-9 receptors via the ERK and NF-kB signaling pathways. HSP also activates CD4+ Th cells, causing the release of IL-2 and further stimulating CTL development, and potentiates the capacity of IL-1 to induce T-cell proliferation. The cytotoxic functions of CTLs, such as granzyme B, perforin, IFN-γ, FasL (CD95L), and TNF-⍺, are elevated by induced hyperthermia, as is Fas receptor presentation on tumor cells.

**Figure 5 cancers-17-03824-f005:**
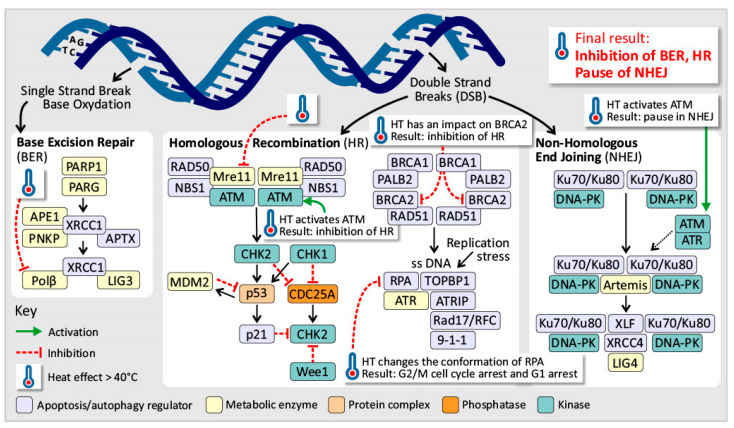
Induced hyperthermia disrupts the DNA repair mechanisms (BER, HR, and NHEJ). At temperatures above 40 °C, induced hyperthermia disrupts DNA damage response pathways, leading to single- and double-strand breaks, impaired base excision repair, homologous recombination and nonhomologous end joining, and ultimately cell cycle arrest or apoptosis.

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
