# Peer review of "Temperature-Dependent Effects of Induced Hyperthermia, Including Whole-Body Hyperthermia, on the Hallmarks of Cancer: A Systematic Review"

_cancers, 2025, doi:10.3390/cancers17233824_

Round 1

Reviewer 1 Report

Comments and Suggestions for Authors

The authors have provided an extensive and comprehensive review of the biological effects of hyperthermia. The text is well written en the figures are wel presented and of good quality.

I have three minor remarks

  1. the authors have explicitly searched reviews and meta-analysis only, not primary papers, this should be mentioned as a weakness in the discussion
  2. due to point 1. the authors fail to indicate in which biological system (in vitro, in vivo model system or in vivo human) the effects were observed. could the authors elaborate on this in the discussion?
  3. due to point 1. and 2.  the authors fail to indicate which of all the observed biological mechanisms is most relevant or promising kandidate to pusuit in the clinic/  in patients and what temperatures we should aim for. Can the authors elaborate specifically on in vivo patient data available?

Reviewer 2 Report

Comments and Suggestions for Authors

Dear editor

The manuscript entitled "The Effects of Induced Hyperthermia, Including Whole-body Hyperthermia, on the Hallmarks of Cancer: A Temperature-Dependent Systematic Review" discusses about temperature-based therapies which can be safely combined with other treatments of cancer.This manuscript can be considered for publication after major revision and addressing following comments point by point.

1- Subclassification of introdution is not usaual. They should be merged 

2- The conclusion is poorly written. It should be completed 

3- Why was PubMed just used as databases?

4- If figures of literature are used, the permission should be given according to the publisher copyright and citation in figures captions.

5- Add a section critically discusses about literature reviewed in present manuscript 

6- It can be suggested to add a table includes summarized data of literature reviewed in present study.

7- Authors should change subject of study . "A Temperature-Dependent Systematic Review" is not usaual. Temperature is not type of review!

Round 2

Reviewer 2 Report

Comments and Suggestions for Authors

The citations should be added , as last column in the right, to Table provided in revise. 
